# Cortical signatures of precision grip force control in children, adolescents, and adults

**Mikkel Malling Beck[1]\*, Meaghan Elizabeth Spedden[1], Martin Jensen Dietz[2], Anke Ninija Karabanov[1,3], Mark Schram Christensen[4], Jesper Lundbye-Jensen[1,4]**

[1]Department of Nutrition, Exercise and Sports (NEXS), University of Copenhagen, Copenhagen, Denmark; [2]Center of Functionally Integrative Neuroscience, Institute of Clinical Medicine, Aarhus University, Aarhus, Denmark; [3]Danish Research Centre for Magnetic Resonance (DRCMR), Hvidovre Hospital, Hvidovre, Denmark; [4]Department of Neuroscience, University of Copenhagen, Copenhagen, Denmark

**Abstract** Human dexterous motor control improves from childhood to adulthood, but little is known about the changes in cortico-cortical communication that support such ontogenetic refinement of motor skills. To investigate age-related differences in connectivity between cortical regions involved in dexterous control, we analyzed electroencephalographic data from 88 individuals (range 8-30 years) performing a visually guided precision grip task using dynamic causal modelling and parametric empirical Bayes. Our results demonstrate that bidirectional coupling in a canonical 'grasping network' is associated with precision grip performance across age groups. We further demonstrate greater backward coupling from higher-order to lower-order sensorimotor regions from late adolescence in addition to differential associations between connectivity strength in a premotor-prefrontal network and motor performance for different age groups. We interpret these findings as reflecting greater use of top-down and executive control processes with development. These results expand our understanding of the cortical mechanisms that support dexterous abilities through development.

**\*For correspondence:** mib@nexs.ku.dk

**Competing interests:** The authors declare that no competing interests exist.

## Introduction

The ability to grasp and manipulate objects is a prominent feature of the human movement repertoire, and the refinement of skilled manual motor control represents a cornerstone of human ontogenetic development. Dexterous abilities and fine manual control improve from childhood through adolescence to early adulthood (*Dayanidhi et al., 2013*; *Forssberg et al., 1991*), but research focused on understanding the cortical control mechanisms that support age-related changes in manual control is sparse.

In adults, the control of the intrinsic hand muscles during externally guided pinching or grasping engages an extensive cortical network (*Castiello and Begliomini, 2008*; *Olivier et al., 2007*): involvement of the inferior parietal lobule (IPL) alongside both ventral premotor (PMv) and supplementary motor area (SMA) as well as the primary motor cortex (M1) has been demonstrated systematically (*Castiello, 2005*; *Castiello and Begliomini, 2008*; *Davare et al., 2010*; *Karabanov et al., 2019*). Additionally, increased activity in the dorsolateral prefrontal cortex (DLPFC) has also been found during tasks requiring skillful scaling of precision grip forces (*Coombes et al., 2011*; *Spraker et al., 2009*). Studies mapping the functional development of sensorimotor networks from childhood to adulthood are lacking, but patterns of ontogenetic structural brain development are well documented and show both gray matter (*Gogtay et al., 2004*; *Sowell et al., 2003*) and white matter (*Asato et al., 2010*; *Lebel and Beaulieu, 2011*) changes in brain regions implicated in

grasping and manual control. Given that the state of neuroanatomical structures could affect the efficiency of functional neural communication in distributed networks of the brain (*Boorman et al., 2007*; *Hagmann et al., 2010*) and that efficient network communication is involved in orchestrating behavior (*Luna et al., 2015*), it seems reasonable to speculate that age-related differences in the ability to perform visually guided motor tasks could also partly rely on a refinement and maturation of cortical processing.

Some studies have investigated the development of cortical control mechanisms supporting hand and finger movements during simple, cued motor tasks and demonstrated age-related differences in spectral features of the electro- or magnetoencephalogram (E/MEG). Specifically, movement-related modulation of oscillatory activity in primary sensorimotor regions was found to be lower in children compared to adults (*Heinrichs-Graham et al., 2018*; *Trevarrow et al., 2019*; *Wilson et al., 2010*). These studies provide important knowledge on age-related differences in motor control processes in core sensorimotor brain areas (*Gaetz et al., 2010*; *Halder et al., 2007*; *Johnson et al., 2019*; *Trevarrow et al., 2019*), but do not allow an assessment of the functional relevance of these differences to skilled motor control as they measure brain activity during the preparation and execution of simple motor tasks without a formal requirement for the quality of the motor output, for example, precision or speed (*Trevarrow et al., 2019*; *Wilson et al., 2010*). Additionally, these studies have focused on regional developmental differences, but have not investigated age-related differences in interregional brain connectivity. Resting-state fMRI studies suggest that connectivity patterns between core sensorimotor areas mature ontogenetically early, whereas executive and attentional networks reach adult-like properties later during development (*Grayson and Fair, 2017*). However, the behavioral relevance of developmental changes in brain connectivity is not known due to the lack of studies relating task-related connectivity in sensorimotor networks to behavioral measures of motor performance.

Communication between different brain regions can be inferred using dynamic causal modeling (DCM) of electrophysiological data. DCM uses a biologically informed generative model of electrophysiological data to make inferences about the effective connectivity between brain areas. Previous studies have utilized this approach to investigate active motor control processes in young adults (*Bönstrup et al., 2016*; *Herz et al., 2012*), non-pathological aging (*Michely et al., 2018*; *Spedden et al., 2020*), and in individuals presenting with neurological conditions such as stroke (*Grefkes et al., 2008*; *Larsen et al., 2018*) and Parkinson's disease (*Herz et al., 2014*). Furthermore, the recent development of refined analytical approaches such as the parametric empirical Bayes (PEB) (*Friston et al., 2016*) enables one to relate estimates of effective brain connectivity to behaviorally relevant outcome measures such as precision of motor output (*Zeidman et al., 2019*).

In the present study, we used DCM for cross-spectral densities (CSDs) to investigate age-related differences in cortical connectivity and their relevance for fine manual control in children, adolescents, and young adults. We did so by measuring EEG activity in 88 healthy individuals from four different age groups while they performed a visually guided isotonic precision grip force-tracing task (*Figure 1*). Specifically, we were interested in effective connectivity that mediates communication between cortical areas in a uni-hemispheric extended sensorimotor network (*Figure 1*). We focused on the contralateral hemisphere as this has most consistently been demonstrated to be involved in the control of the precision grip (*Cavina-Pratesi et al., 2018*; *Culham et al., 2006*; *Olivier et al., 2007*). Using PEB (*Friston et al., 2016*; *Zeidman et al., 2019*), we asked which cortical connections best explained shared commonalities between individuals and which best explained differences due to precision grip performance, due to age group independent of task performance and their interaction. This approach allowed us to investigate cortical connectivity that was related to motor performance and connectivity that differed between individuals at different stages of typical development, but was not necessarily task-dependent. We expected that differences in effective connectivity within this extended sensorimotor network would be associated with precision grip performance independent of age group; that patterns of connectivity would differ between age groups independent of task performance; and finally, that some coupling parameters would display distinct associations to motor performance for different age groups.

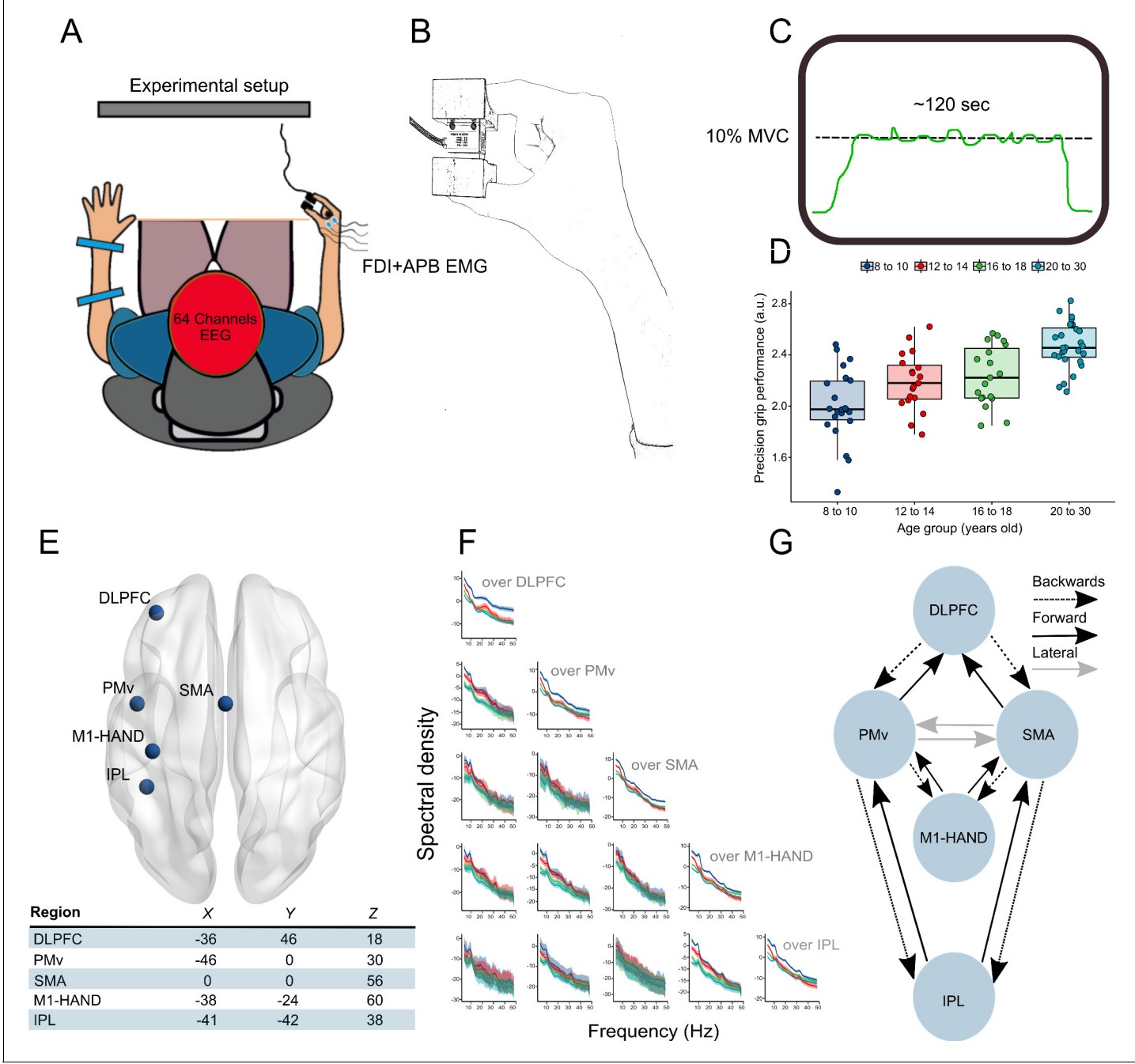

**Figure 1.** Experimental setup and framework of analysis. (**A**) Participants were seated in front of a screen, and 64-channel EEG and EMG from the first dorsal interosseous (FDI) and abductor pollicis brevis (APB) was acquired. (**B**) Precision grip force transducer and hand configuration during the task. Part (**C**) displays the precision grip force-tracing task. A horizontal target line representing 10% of maximal voluntary contraction was centered in the middle of the screen, and participants were instructed to track this line as closely as possible by adjusting the force exerted between their index finger and thumb. (**D**) Boxplots representing performance in the precision grip task for the different age groups. (**E**) Regions shown to be involved in visually guided precision grip tasks and the corresponding MNI coordinates in mm used for the DCM analysis. (**F**) Auto and crossspectral densities for sensors located above the brain regions of interest. Solid lines represent group means, and shaded ribbons display 95% CIs. (**G**) Schematic representation of the extrinsic connections between brain regions specified in the DCMs.

The online version of this article includes the following source data and source code for figure 1:

**Source code 1.** R-script used to analyze and plot motor performance data presented in *Figure 1D*.
**Source data 1.** Motor performance data for behavioral analysis.

## Results

We recruited 98 healthy volunteers, but excluded 10 individuals due to excessive muscle and movement artifacts in the EEG (n = 4), an inability to comprehend and/or perform the task (n = 5), and technical malfunctions (n = 1). This resulted in a sample of 88 individuals from four age groups: 8–10 years (n = 22), 10–12 years (n = 19), 16–18 years (n = 19), and 20–30 years (n = 28) (*Table 1*).

### Fine manual control is better in older individuals

To investigate differences in manual control, participants were asked to perform a force-tracing task requiring fine adjustments of the force exerted between the index finger and thumb in a precision grip to track a horizontal target line (*Figure 1A–C*). Precision grip performance was significantly different between groups, as revealed by a one-way ANOVA demonstrating a significant main effect of age group ($F_{84,3}$ = 17.8, p<0.001). Post-hoc comparisons using two-sided t-tests adjusted for multiplicity using the Holm-adjustment demonstrated that 8–10-year-old individuals performed the task with significantly lower precision compared to the 12–14-year-old ($df$ = 40, p=0.019), the 16–18-year-old ($df$ = 40, p=0.005), and the 20–30-year-old ($df$ = 49, p<0.001). Additionally, the 20–30-year-old performed the task with significantly higher precision compared to the 12–14-year-old ($df$ = 46, p<0.001) and the 16–18-year-old ($df$ = 46, p=0.003) (*Figure 1D*). No significant differences were found in precision performance between the 12–14-year-old and the 16–18-year-old ($df$ = 37, p=0.55).

### Spectral features in children, adolescents, and adults that were captured by fitted DCMs

We were interested in investigating developmental differences in connectivity between cortical areas implicated in fine manual control. We focused on an extended sensorimotor network comprising contralateral parietal, primary motor, premotor, and prefrontal regions (*Cavina-Pratesi et al., 2018*; *Culham et al., 2006*; *Olivier et al., 2007*). Group-averaged auto- and cross-spectral densities on the sensor level are presented in *Figure 1F*. Clear spectral features were observed across groups in alpha, beta, and gamma bands in the auto- (diagonals) and cross-spectra. In accordance with previous studies using resting-state measurements (*Cragg et al., 2011*; *Miskovic et al., 2015*; *Rodrí-guez-Martínez et al., 2015*), we observed that, independent of frequency, absolute regional power was generally greater across the included electrodes in children compared to late adolescents and young adults (*Figure 1F*). This was generally also the case for the CSDs. These differences in scalp data dynamics due to age group or task performance encouraged our DCM analysis (*Figure 1G*). We fitted one full DCM consisting of directed reciprocal connections between five a priori chosen key brain regions supporting manual control processes to the EEG data from each participant. The sources included in the DCM were the IPL, M1, PMv, SMA, as well as the DLPFC (*Figure 1E*). These sources were chosen a priori based on previous imaging literature investigating brain activity during precision grip motor tasks (*Cavina-Pratesi et al., 2018*; *Ehrsson et al., 2001*; *Kuhtz-Buschbeck et al., 2001*) and because they display distinct developmental trajectories (*Shaw et al., 2008*). The specified directed connections between regions followed principles of functional hierarchical cortical processing (*Felleman and Van Essen, 1991*). This entailed reciprocal connections between the regions of interest with backward (top-down) connections projecting from higher-order to lower-order regions and forward (bottom-up) connections projecting from lower-order to higher-order regions (*Felleman and Van Essen, 1991*; *Shipp et al., 2013*; *Figure 1G*). Observed and predicted auto- and cross-spectra from the first four principal modes (or components) of EEG data from

**Table 1.** Participant characteristics.

|  | 8–10 years | 12–14 years | 16–18 years | 20–30 years |
| --- | --- | --- | --- | --- |
| Number of participants | 22 | 19 | 19 | 28 |
| Sex (F/M) | 9/13 | 9/10 | 10/9 | 16/12 |
| Age (months) | 108 ± 7 | 156 ± 7 | 206 ± 7 | 301 ± 34 |
| Tanner developmental stage | 1.12 ± 0.27 | 2.56 ± 1.03 | 4.63 ± 0.55 | 5 ± 0 |

Data presented as counts or as means ± standard deviations (SDs).

an exemplary participant (in the 8–10-year-old age group) are depicted in *Figure 2*. Examples of fitted and observed data from participants in the remaining age groups can be found as a supplement to *Figure 2* (*Figure 2—figure supplement 1*). Notably, the fitted DCMs were able to accurately capture prominent features of the observed data for individuals across all age groups.

Mean variance explained ($R^2$) by the individual DCMs was 96.0 ± 13.3%. No significant differences in model fits (quantified as variance explained) were found between age groups from a one-way ANOVA ($F_{(84,3)} = 0.95$, p=0.42).

## Probing network of included cortical sources

Before moving on to estimating age-related differences in effective brain connectivity between individuals on the group level, we set out to verify that the sources included in the DCM were sufficiently spaced as volume conduction can affect measures of cortico-cortical connectivity from EEG data (*Schoffelen and Gross, 2009*). For this purpose, we defined a second DCM and inverted it on EEG data from each participant. This left us with two inverted DCMs per subject: one full DCM as described in *Figure 1E, G* and *Figure 3* (full DCM) and another DCM (disconnected DCM) in which the IPL was modeled as a hidden source (i.e., without extrinsic connections originating and targeting this source) (*Figure 3*). The evidence for these models given the data was compared using Bayesian model selection (BMS). The rationale behind this analysis was that if a source contributed uniquely to explaining the observed auto- and cross-spectra, then this would *not* be heavily influenced by field spread. This would be reflected as a difference in model evidence between the two models favoring the fully connected model over the disconnected model (full DCM > disconnected DCM). IPL was chosen as the source of interest to be modeled for the given question as this represented a source in close proximity to another (i.e., M1) in the a priori defined network of interest (note the MNI coordinates, see *Figure 1* and *Figure 3*). Model comparison was performed by means of random effects (RFX) BMS. The results of this analysis showed that the model evidence was greater for the full model compared to the disconnected model (*Figure 3*). Exceedance probability for the *full DCM*

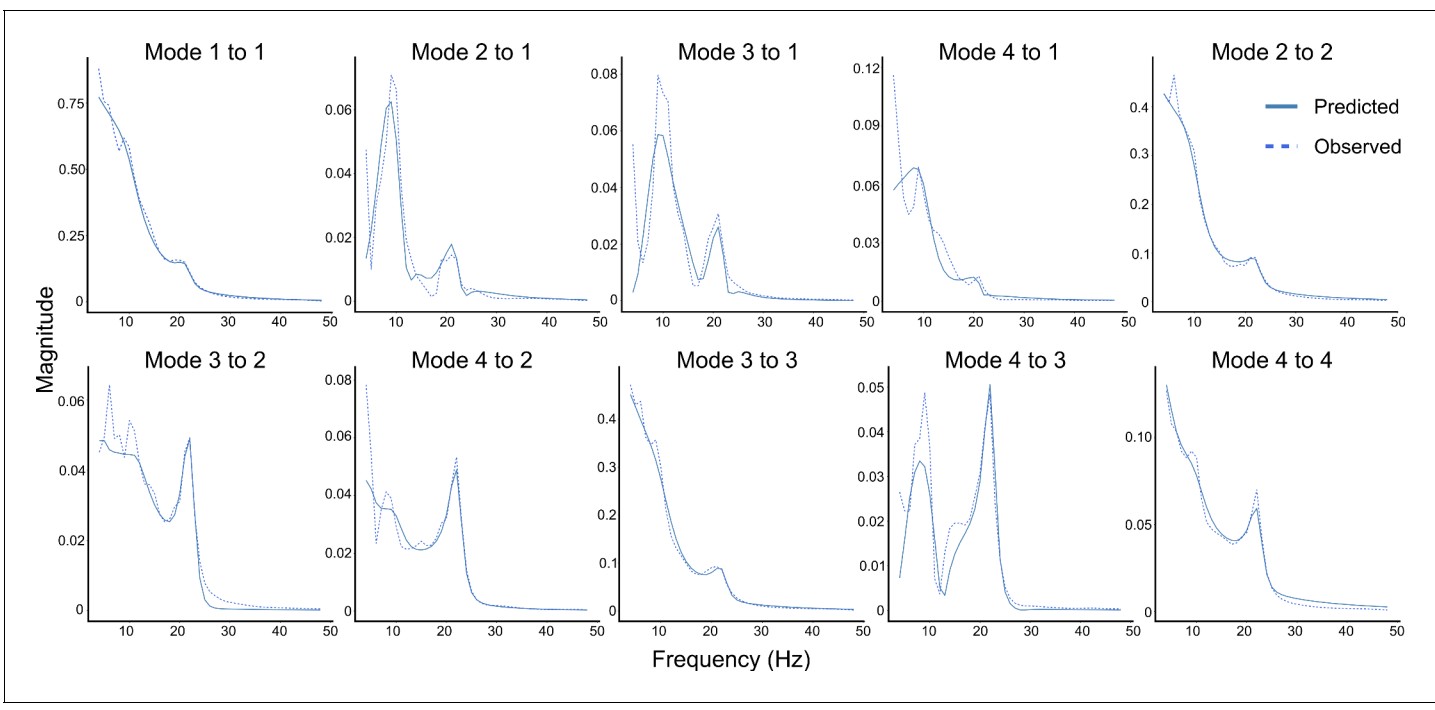

**Figure 2.** Observed and DCM-predicted auto- and cross-spectra. Predicted and observed auto- and cross-spectra for the first four modes from an exemplary participant in the 8–10-year-old age group. See *Figure 2—figure supplement 1* for exemplary data from individuals across all four age groups.

The online version of this article includes the following figure supplement(s) for figure 2:

**Figure supplement 1.** Examples of observed and predicted auto- and cross-spectra from all age groups.

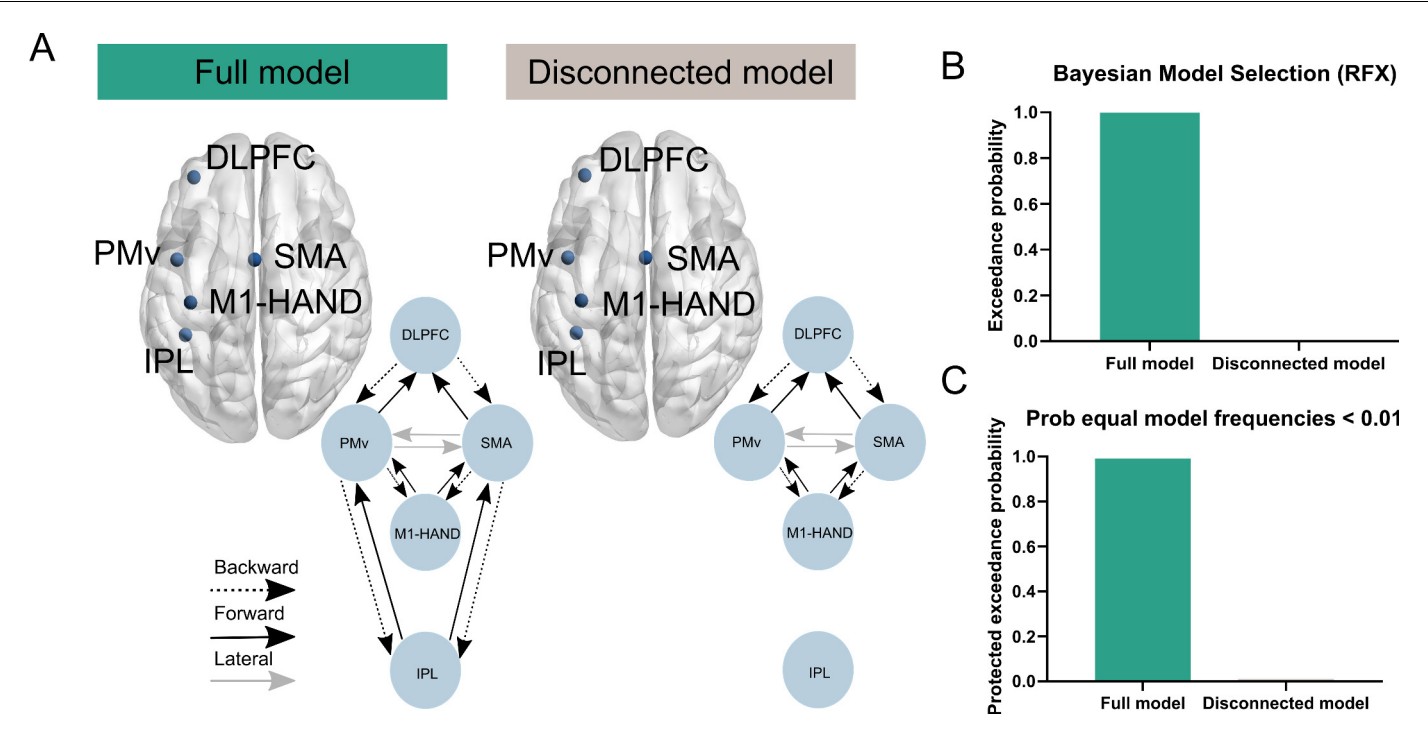

**Figure 3.** Model comparisons. To ensure that the sources included in our a priori DCM were sufficiently spaced to minimize the influence of volume conduction, we performed a random effect (RFX) Bayesian model selection (BMS). (**A**) We compared two different DCMs. The *full model* was our a priori defined model of the contralateral grasping network, whereas the *disconnected model* included the same sources and thereby same data (via principal modes of the EEG data), but the IPL was modelled as a hidden source. This enabled a comparison of the two models using RFX BMS. The model comparison revealed greater evidence for the fully connected model as reflected in a greater exceedance probability and protected exceedance probability (**B, C**).

was 1, whereas that of the *disconnected DCM* was 0. Similar results were found for the protected exceedance probability (full DCM > 0.99; disconnected DCM < 0.01). That is, the full model proved a more parsimonious fit to the principal modes of EEG data under the free-energy approximation to the Bayesian model evidence. It therefore seems likely that each of the sources included in our DCM contributed uniquely to explaining the observed EEG data features.

As a next step, we took the full DCMs from each subject to the group level (second level). Here, we quantified mean connectivity across participants (commonalities); differences in connectivity due to precision grip performance; age group and interactions between performance and age group using PEB.

## Group-level analysis of mean connectivity coupling using PEB

We used an iterative Bayesian model reduction (BMR) scheme to prune away connections that did not contribute positively to model evidence by searching through combinations of model parameters nested in the full model space. Next, a weighted average of the coupling parameters was computed from the final BMR iteration (i.e., Bayesian model average [BMA]). *Figure 4* depicts the estimated connection strengths as log-scaling values in relation to the priors and their uncertainties after thresholding at 95% posterior probability. The average coupling strength across all participants for each of the modeled connections is reflected in the commonalities of the BMA analysis (*Figure 4A*). This shows that all coupling parameters were conserved across participants (posterior probability >0.95), except for the lateral connection from SMA to PMv (posterior probability of 0.88). The negative sign of the estimates reflects that estimates were generally lower than the prior values.

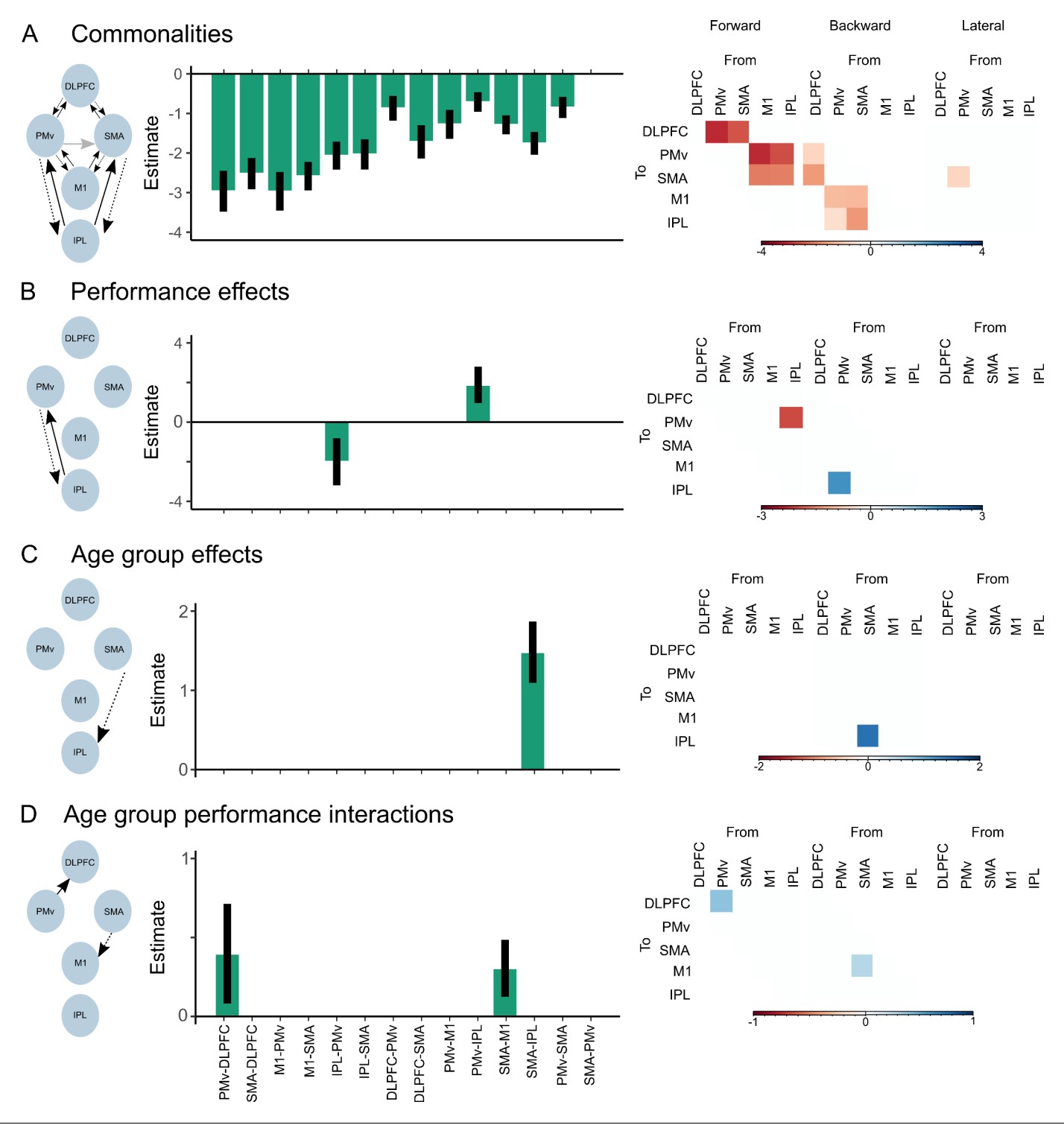

**Figure 4.** Parameters of the group-level parametric empirical Bayes following Bayesian model averaging (BMA). Depicted are only those extrinsic connections surviving the 95% posterior probability threshold based on free energy. Left shows the schematic network presentation. Middle displays the posterior parameter estimates with the associated uncertainties. Right shows the same estimated parameters split into a matrix representing the forward, backward, and lateral extrinsic connections. (**A**) Commonalities (mean connectivity) across participants. (**B**) Differences in connectivity due to precision grip performance. (**C**) Differences in connectivity due to age group. (**D**) Differences in connectivity due to age group vs. precision grip performance interactions.

The online version of this article includes the following source data and source code for figure 4:

**Source code 1.** R-script used to plot results from group-based parametric empirical Bayes analysis presented in *Figure 4*.

**Source data 1.** Group-based parametric empirical Bayes results.

## Bidirectional coupling in a fronto-parietal network is associated with precision grip performance

We found effects of performance on two extrinsic connections in our network, namely the forward connection from IPL to PMv (posterior probability = 1) and the reciprocal backward connection from PMv to IPL (posterior probability = 1) (*Figure 4B*). The negative sign of the IPL to PMv parameter indicates that stronger coupling was negatively associated with precision grip performance, whereas the positive sign of the PMv to IPL estimate suggests that larger PMv-IPL coupling strength was associated with better performance. This was clearer when plotting the relationship between individual estimates of coupling strength and pinch precision performance (*Figure 5A, B*). Here, a clear positive association with performance was found for PMv to IPL coupling, and a negative association was found between IPL to PMv coupling strength and performance. Notably, these relationships were consistently present across different age groups as can be seen from the group-wise Pearson correlation coefficients presented in *Figure 5A, B*.

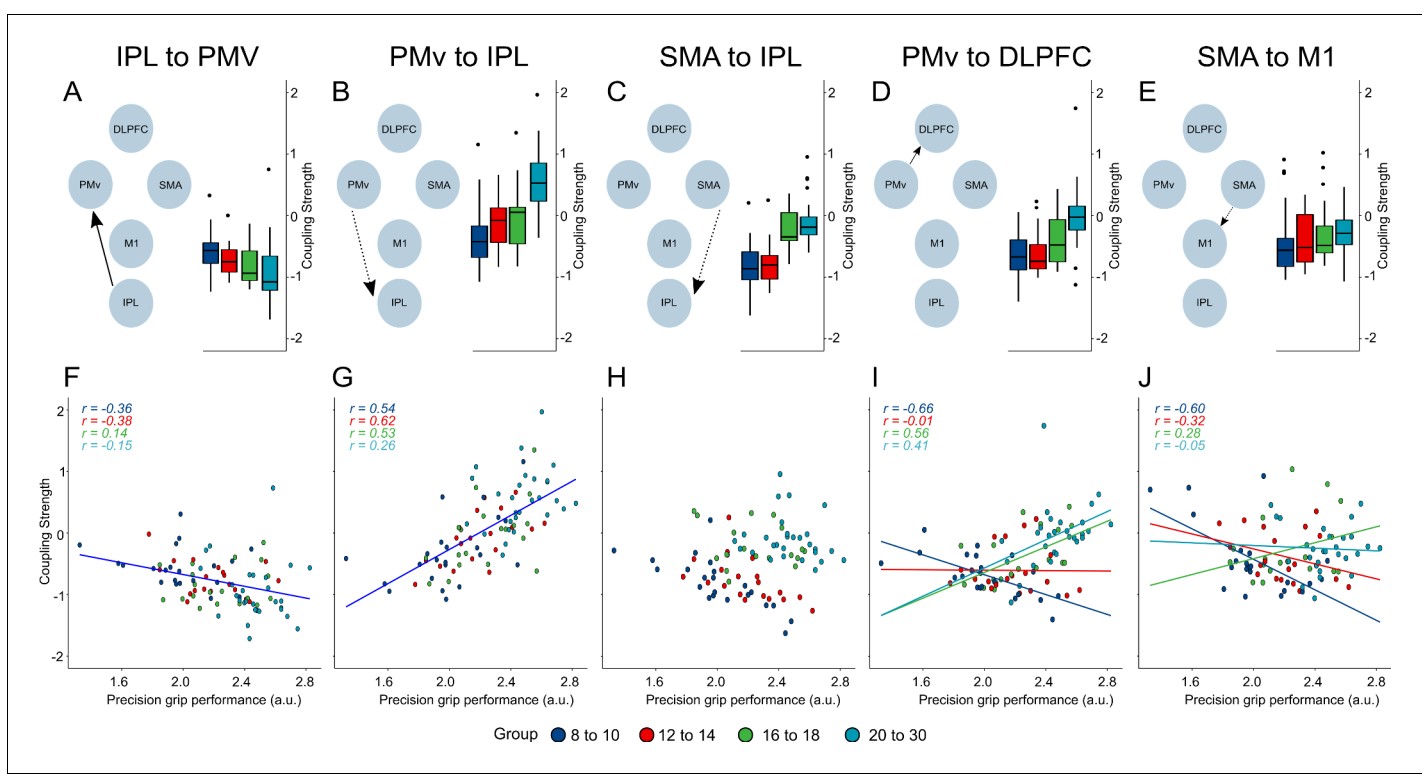

**Figure 5.** Main effects and interaction effects. Schematic representing model for main effects and interaction effects (A–E). Boxplots representing the distribution of coupling strength faceted by age group for the respective connectivity parameter (A–E). Scatter plots (F–J) represent individual data points with one regression line for main effects and regression lines by age groups for interaction effects. Parts (A), (B), (F), and (G) display the association between precision grip performance and coupling strength. (C) and (H) depict the main effect of age group on coupling parameters. (D), (E), (I), and (J) display interactions between age group and performance. Coupling strength values are depicted as unit-less log-scaling entities. The online version of this article includes the following source data and source code for figure 5:

**Source code 1.** R-script used to plot associations between individual coupling parameters and motor performance presented in *Figure 5*.
**Source data 1.** Individual coupling parameters and motor performance.

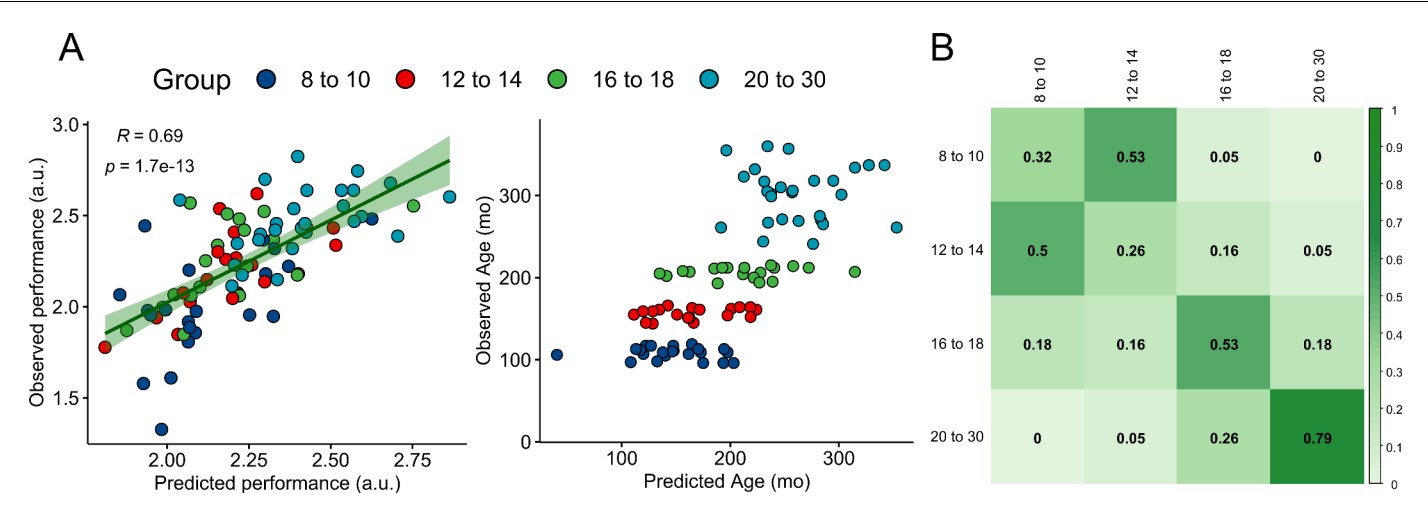

**Figure 6.** Actual and predicted precision grip performance and age. (A) Associations between actual and predicted precision grip performance (left) and fractionized age (right) from leave-one-out cross-validations using multiple linear regression. (B) Confusion matrix normalized to group sample size representing the proportion of predicted labels based on a quadratic discriminant analysis in a LOO routine. The diagonal represents the proportion of correctly identified labels.

The online version of this article includes the following source code for figure 6:

**Source code 1.** R-script used to run and plot cross-validation and classification analyses presented in *Figure 6*.

## Greater backward coupling from SMA to M1 from late adolescence

We observed a main effect of age group on the backward connection from SMA to IPL (posterior probability = 1) (*Figure 4C*). *Figure 5C* shows that individuals older than 16 years seem to display a greater degree of backward coupling compared to individuals younger than 14 years.

## Associations between coupling strength and performance that differ between age groups

The forward connection from PMv to DLPFC displayed an interaction between age and group performance (posterior probability = 1) (*Figure 4D*). The effect is depicted in *Figure 5D and I*. This shows distinct relationships between coupling strength and performance between age groups. For the youngest groups (8–10 years), there was a negative association. In the 12–14 year olds, there was no clear association with performance, evident from the largely flat relationship. For the two oldest age groups (16–18 and 20–30 years), a positive association was present, suggesting that individuals within these age groups with larger PMv-DLPFC coupling strength performed the task with higher precision. The coupling from SMA to M1 also displayed an age group by performance interaction (posterior probability = 1) (*Figure 4D*). This effect is depicted in *Figure 5E and J*. Here, we observed a negative association with performance in the 8–10 year olds and the 12–14 years olds. In contrast, this association was positive or largely flat for individuals aged 16–18 years and the young adults, respectively.

## Predicting participant age and performance from coupling strengths

Lastly, we tested the generalizability and predictive validity of our results utilizing a leave-one-out cross-validation (LOOCV) routine. Multiple linear regressions containing the updated and extracted coupling strength parameters as covariates and participants' fractionized age (operationalized continuously in months) and precision grip performance as the dependent variable were performed in two separate analyses. The relationships between observed and predicted values of performance and age are presented in *Figure 6A*. For precision grip performance, a multivariate regression model was found to predict out-of-sample motor performance (root mean squared error [RMSE] = 0.21 a.u.) accounting for 47% of the variance ($R^2 = 0.47$) (*Figure 6A*, left). For continuous age in months, multiple linear regression found predictive value (RMSE = 53.4 months) for

extrinsic coupling parameters that accounted for 54% of the variance ($R^2$ = 0.54) (*Figure 6A*, right). We further used quadratic discriminant analysis (QDA) as a multiclass classifier to predictively label participants' age group (as a categorical variable) from the coupling strength estimates. *Figure 6B* displays the normalized confusion matrix representing the proportion of identified age group labels for each age group (i.e., the sensitivity is expressed on the diagonal). Classifier accuracy was 50% (CI [0.39 0.60]), and this was significantly better than a naïve classifier designed to draw the most common age group label (p=0.002).

## Discussion

This study used biophysically informed models of EEG spectral data to investigate associations between estimates of brain connectivity, age, and fine manual control in a large sample of individuals at different stages of typical development. We found that bidirectional effective connectivity between premotor and parietal regions was associated with precision grip performance across individuals. Notably, effective coupling from PMv to IPL was positively associated with performance, whereas connectivity strength from IPL to PMv was negatively related to performance. These results support a prominent role for communication in a network comprising PMv and IPL in precision grip control and suggest that larger top-down connectivity within this canonical grasping network, as opposed to bottom-up connectivity, positively supports fine motor performance across individuals at different stages of development. Additionally, independent of performance, individuals aged 16 and above expressed a greater degree of backward coupling between supplementary motor and parietal regions while a positive association between premotor and prefrontal coupling strength and performance was also found in these individuals. This potentially reflects a greater degree of top-down control and use of additional executive control processes from late adolescence and onwards compared to childhood. Below, we discuss these patterns of task-based cortical connectivity. We focus on connectivity estimates that are related to performance across individuals independent of age; those that differ due to age group, but are not specifically related to motor performance; and those that relate to performance distinctively as a function of age.

### Coupling within a canonical grasping network support performance in children, adolescents, and adults

Frontal and parietal cortical regions are recognized as part of the core of the brain's 'grasping' network (*Binkofski et al., 1999*; *Castiello, 2005*; *Castiello and Begliomini, 2008*; *Fogassi et al., 2001*; *Gallese et al., 1994*). A prominent theory is that premotor and parietal areas continuously exchange information through reciprocal functional connections, thus forming a recurrent loop that subtends control of grasping behavior (*Castiello and Begliomini, 2008*; *Davare et al., 2011*). Our data provide empirical support for this view by demonstrating that bidirectional coupling between premotor and parietal regions is related to the accuracy with which one controls visually guided pinch force. Additionally, our results revealed dissociated relationships for the directed influences between parieto-frontal areas on precision grip performance. Specifically, backward coupling from PMv to IPL was positively related to performance, whereas forward coupling from IPL to PMv was negatively associated with performance, indicating that individuals that performed well displayed high backward PMv-IPL or low forward IPL-PMv coupling. These results may be understood from the theory of active inference in the sensorimotor system (*Adams et al., 2013*; *Friston, 2010*). Conceptually, it has been proposed that backward (top-down) connections relay predictions about sensory consequences of actions, whereas forward (bottom-up) connections convey feedback information on the differences between the actual and intended actions (prediction errors) (*Adams et al., 2013*; *Bastos et al., 2012*). In the context of the functional hierarchical architecture of the parieto-frontal network specified here, this would entail that premotor areas provide parietal areas with predictions of sensory consequences of planned actions. Parietal areas reciprocate by furnishing premotor areas with bottom-up sensory information relating to the performed movement; and the mismatch between the intended and the actual movements that may be used for empirical updating through ongoing reciprocal exchange of information (*Christensen et al., 2007*; *Davare et al., 2011*; *Jenmalm et al., 2006*; *Tunik et al., 2005*). Drawing on this framework, we argue that the results of the present study provide evidence that effective use of cortical control mechanisms within a parieto-frontal 'grasping network' supports fine force regulation during precision grip, and that

differences in performance between individuals might be partly explained by the very same mechanisms. Individuals that perform well are characterized by a greater degree of (accurate) predictive, top-down control compared to poorer performing individuals. We further observed that forward coupling from IPL to PMv was negatively associated with precision grip performance. The negative association to performance for this forward connection could likely arise from suboptimal predictive control mechanisms itself. Suboptimal predictive control would demand more feedback for online adjustments and updating in the form of ascending information from parietal to premotor areas based on, for example, visual information. This would be mediated by forward connections when precision grip performance deviates from the expected or when predictions are not well tuned (i.e., larger prediction errors). The connectivity-performance relationships that favor high backward coupling seem likely for the task used in the current study. One would indeed expect that adopting a predictive control strategy could be beneficial for task performance given the stationary target. Whether the same control mechanisms are similarly involved in other precision grip tasks needs to be elucidated, although one recent study found patterns of functional connectivity that could support these views (*Iturrate et al., 2018*).

Considering the parieto-frontal connectivity patterns from a developmental perspective, it is particularly interesting that these relationships were present across different age groups from 8 to 10 years of age and to adulthood. This further solidifies the central role of fronto-parietal networks in mediating control of precision grip and grasping. More importantly, this also suggests that this network is functionally available already from mid-to-late childhood (i.e., from 8 years). This is in line with earlier findings suggesting that the emergence of behavioral indices reflecting predictive control strategies of precision grip force coordination can be seen in children aged 8 years (*Forssberg et al., 1991*) and is also further strengthened by the fact that other sensorimotor networks supporting proprioceptive function seem to display similar developmental properties (*Cignetti et al., 2017*). The early structural maturation of core sensorimotor cortical areas (*Gogtay et al., 2004*; *Shaw et al., 2008*) could perhaps provide the scaffold that support basic functional network interactions enabling certain skill levels of sensorimotor functions relatively early during development. That being said, precision grip performance was found to be lower in children aged 8–10 compared to older individuals in the present study. Furthermore, individuals younger than 18 years were outperformed by adults. This suggests a later ontogenetic trajectory for other control mechanisms within neural systems aside from canonical grasping networks that could be age-dependent and potentially relevant for motor functioning.

## Late adolescence and early adulthood are characterized by greater top-down control

The results of the present study indicate that backward connectivity from SMA to IPL was positively related with age. Given that backward connections mediate top-down control, this result suggests that, independent of task performance, individuals aged 16 and above display a larger degree of top-down control. Additionally, for the age group-performance interactions, the estimated backward coupling from SMA to M1 only followed a clear negative association with performance for individuals younger than 14, possibly reflecting an inability to capitalize on efficient top-down control to guide precision grip performance for younger individuals. This was not the case for the individuals older than 16. This suggestion of a greater degree of efficient top-down control with advancing age harmonizes well with a previous study from our group investigating oscillatory coupling within the corticospinal system. Using directionality analyses of corticomuscular coherence, we demonstrated a larger proportion of descending (top-down), relative to ascending (bottom-up), oscillatory coupling as a function of age (in 7–23-year-old participants) during a force-tracing task using the ankle muscles (*Spedden et al., 2019*). Echoing the results from the current study, the greater proportion of descending relative to ascending oscillatory coupling could support the notion of an increased reliance on predictive, top-down control with age. These age-related differences could arise as the nervous system fine-tunes its predictions of the sensory consequences of its enforced actions through repeated 'learning' experiences (and empirical updating of beliefs) during development (*Moran et al., 2014*). As such, even though the structural framework and the overall (resting-state) connectivity profiles are organized ontogenetically early in sensorimotor systems (*Grayson and Fair, 2017*; *Lebel and Beaulieu, 2011*; *Shaw et al., 2008*), changes in the balance or efficiency of communication within cortical networks may still occur with development as shown in the present study.

Indeed, our results suggest that some degree of refinement of directed functional interactions between sensorimotor cortical regions extends into adolescence reaching adult-like properties late during adolescence. Although analyses were based on EEG recordings that were obtained while participants performed a precision grip task, these specific developmental differences in cortical connectivity were not related to precision grip performance per se as revealed by the PEB analysis. The functional relevance of the reported age-related differences in coupling may therefore not be specific to the development of skilled grasping. Instead, they could reflect a general reorganization of directed brain connectivity patterns that may also be seen for other behavioral states or tasks. This also resonates well with previous findings suggesting a shift from bottom-up to top-down processing during cognitive tasks (for example, taxing response inhibition) as children and adolescents approach adulthood (*Bitan et al., 2006*; *Hwang et al., 2010*). Our results thus extend these trajectories observed in classical cognitive tasks to the domain of precision motor control of the hand. This parallels the developmental trajectories known from structural neuroimaging studies: brain areas and white matter structures supporting top-down control processes develop later than those specialized in processing and conveying sensory-motor information (*Asato et al., 2010*; *Lebel and Beaulieu, 2011*; *Shaw et al., 2008*). Collectively, this suggests that a shift towards a greater reliance on top-down processing at the expense of bottom-up processing with advancing age may reflect a common neurodevelopmental hallmark for functional brain interactions. The design of the study did not include a control task or state (for example., a recording during rest), and we therefore cannot make direct comparisons between task-specific and general differences in connectivity across ages. This also means that the observed age-related differences could be influenced by task-irrelevant factors such as the signal-to-noise ratio and how well the DCMs captured the oscillatory patterns of the obtained EEG signals. That being said, our primary interpretations are based on estimated connectivity obtained through inversion of DCMs, and we did not observe statistical differences in how well the estimated DCMs explained the variance of the EEG data features between age groups.

## Taking advantages of executive processes for motor control? Associations between prefrontal-premotor coupling and precision grip performance are distinct for different age groups

The protracted developmental trajectory of higher-order prefrontal structures in combination with the neuroanatomical connections between prefrontal and non-primary motor areas positions information flow to and from prefrontal areas at the apex of hypothesized regions through which age-related differences in cortical connectivity could affect motor performance. Our results suggest that the association between PMv to DLPFC coupling and precision grip performance was indeed age group dependent. Relationships between PMv-DLPFC connectivity and motor performance scaled differentially for different age groups. A negative association was found for the group of 8–10 year olds, whereas a shift towards positive associations gradually emerged in the 12–14 year olds. This association was clearly positive for the individuals aged 16 and above. Altogether, this suggests that patterns of directed premotor-prefrontal connectivity exhibit distinct age-related profiles that are differentially related to motor behavior. These findings could represent gradually developing adaptations of premotor-prefrontal functional interactions that support greater manual force control. We speculate that the increased information flow to the DLPFC with age could reflect a partial shift in the 'strategy' adopted by the central nervous system to perform the task for individuals at different developmental stages. Whereas young children rely on canonical sensorimotor systems to guide behavior, individuals in late adolescence and early adulthood (aged 16 and above) have the additional possibility to draw on higher-order systems mediating executive control processes. Although speculative, this increased coupling targeting the DLPFC could perhaps be involved in executive control and online error or performance monitoring (*Segalowitz et al., 2010*; *Tamnes et al., 2013*), with beneficial implications for motor precision in developed executive systems (*Friston, 2010*). It could also be related to modulation of 'attention towards action' involving the prefrontal cortex (*Rowe et al., 2002*), although such modulatory effect of attention-to-action would perhaps be more consistent with the physiological mechanisms exerted by backward connections. Independent of its specific role, the notion of positive effects of prefrontal influences on motor behavior is well known from studies investigating motor control mechanisms in healthy aging. Some of these studies have shown that older individuals tax prefrontal structures to a larger degree while performing motor tasks compared to younger adults with positive implications for motor performance

(*Heuninckx et al., 2008*; *Heuninckx et al., 2005*; *Michely et al., 2018*). This pattern can be interpreted as an adaptive compensatory mechanism (*Cabeza et al., 2018*) that serves to attenuate the functional impairments caused by structural and functional declines in core sensorimotor circuitry (*Seidler et al., 2010*). However, it also demonstrates the basic principle that engaging processes within or between brain areas that are commonly ascribed to higher-order cognitive processes (for example the DLPFC) may also benefit motor functions. This notion is supported by our data, but only for those individuals who are at a later stage of development, that is, late adolescence and adulthood, perhaps because these systems and processes mature ontogenetically late.

## Out-of-sample predictions provide putative circuit-based biomarkers of motor function and age

From the current results, we were able to make predictions of left-out participants' skilled motor performance through LOOCVs. Basic patterns of connectivity could also be used to make robust inferences of left-out participants' true age. Collectively, these results suggest that task-based effective connectivity provides markers capable of detecting developmental and motor performance differences. This suggests that brain connectivity measures hold predictive value as putative circuit-based biomarkers of grasping performance in able-bodied individuals of different ages and at different stages of development. It is appealing that analyses of task-based EEG recordings offer predictions of these measures. Whether adopting a similar approach to the one used in this study could be sensitive to differentiate between typically developing individuals and those who are challenged by neurodevelopment disorders affecting cognitive-motor systems may be of interest for future work. That being said, it is also important to acknowledge that interpretations of the present results depend on the specific network model tested. A consequence of this is that models with different connectivity architectures might offer equally – or more – probable explanations of the observed data. Of note, DCMs for electrophysiological data are designed for modeling sparse network structures and do not accommodate large numbers of sources well as this may increase the risk of local minima during model inversion (*Litvak et al., 2019*; *Zeidman et al., 2019*). We chose to fit our DCM to cortical sources consistently implicated in motor and executive control processes, and restricted the network to the contralateral hemisphere, as this has most robustly been implicated in control of the uni-manual precision grip (for example *Cavina-Pratesi et al., 2018*; *Culham et al., 2006*). However, this also entails that we cannot conclude on age-related or performance-specific roles of connectivity between, for example, ipsilateral or subcortical brain regions.

Collectivity, our results provide a characterization of differences in effective brain connectivity within an extended cortical sensorimotor network related to age and performance in a precision grip motor task in children, adolescents, and adults. In doing so, the results expand our knowledge on the cortical control mechanisms that support dexterous motor functions in humans and how performance-specific and general patterns of connectivity develop from childhood to adulthood.

## Materials and methods

### Participants

We recruited 98 healthy participants from four different age groups: 8–10 years (n = 26), 12–14 years (n = 21), 16–18 years (n = 21), and 20–30 years (n = 30). Participants had no known neurological or psychiatric disease nor were they taking medication that could affect central nervous system functioning. Before enrolling in the study, all participants assented to the study procedures, and written informed consent was obtained from participants (>18 years) and their parents (<18 years). The study was approved by the regional ethical committee (H-17019671), and all procedures adhered to the Declaration of Helsinki.

### Precision grip force task

Participants were instructed to perform a force precision task by applying pinch force to a load cell (Dacell, AM210, Dacell Co. Ltd., Korea) held between their index and thumb (i.e., precision grip) (for task setup, see *Figure 1*). The force applied to the load cell was amplified (×100), low-pass filtered (10 Hz), and digitized (1000 Hz) (CED1401, Cambridge Electronic Design Ltd, Cambridge, England) and displayed as a real-time trace representing the applied force on a 30″ monitor placed on a

table ~50 cm in front of the participant at eye level. Also projected on the screen was a fixed horizontal target line corresponding to 10% of MVC for each individual. The y-axis was scaled from 0 N to 20% MVC, and the x-axis was sliding at a steady pace, but fixed at a width of 20 s. Participants were instructed to trace the horizontal target line as precisely as possible for 120 s by adjusting the level of force applied to the load cell. Task performance was quantified as the RMSE between the actual force and the horizontal target line over the 120 s. The RMSE scores were log-transformed to normalize their distribution. The log-transform yielded a negative score where scores that were more negative reflected a smaller RMSE (i.e., less error). Then, the scores were reverse coded (multiplied by −1), giving them a more straightforward interpretation (as pinch precision). As such, higher scores reflect better pinch precision performance. To determine whether precision grip performance significantly differed between groups, we performed a one-way ANOVA with age group as a factor using RStudio (*R Development Core Team, 2019*).

## EEG recordings

Participants' EEG was sampled while they performed the pinch force task. Participants were equipped with an electrode cap comprising 64 active channels placed according to the international 10-20 system (BioSemi, Amsterdam, The Netherlands). EEG was sampled as raw signals using the ActiView software (v 7.07) with a sampling rate of 2048 Hz. According to the BioSemi amplifier design, online referencing is provided by the common mode sense (CMS) and the driven right leg (DRL) electrodes. EEG signals were recorded continuously for 120 s while participants performed the force-tracing task, during which it was visually ensured that the electrode offset was kept $\leq$ 30 µV. Participants were instructed to relax their face and neck to minimize artifacts from muscle activity in the EEG recordings.

## EEG preprocessing

Data preprocessing was carried out using EEGLAB (v. 14.1.1) interfaced in MATLAB (v. 2017b). First, data was band-pass filtered from 0.5 to 48 Hz. Then, the signals were visually inspected to identify channels or time periods displaying excessive noise, and these were subsequently removed from the data. Signals were then re-referenced to average reference before they were subjected to an independent component analysis (ICA). Components reflecting eye blinks and/or horizontal eye movements were identified and removed per visual inspection (*Chaumon et al., 2015*), and removed channels were interpolated using the spherical interpolation scheme embedded in EEGLAB. Please consult *Supplementary file 1* in the supplementary files (*Supplementary file 1*) for a full description of the EEG preprocessing pipeline. To visualize the spectral content of the recorded and preprocessed EEG signals, we computed the power and CSDs for five sensors located over regions implicated in visually guided manual control of force in previous imaging studies, namely DLPFC, PMv, SMA, M1, and IPL, using default Welch's averaged periodogram in MATLAB. The auto- and cross spectral densities are displayed for the frequency range 4–48 Hz in *Figure 1F*.

## DCM of CSDs for subject-level analysis

To infer effective connectivity between cortical regions of interest, we used the DCM framework implemented in SPM12 (v 7487). DCMs model dynamic interactions within and between sources (model parameters) to allow inference of effective connectivity, that is, the effect that one brain area exerts over another. Here, we used DCMs for CSD – made for quasi-stationary neural time series – to explain derived auto- and cross-spectral densities from principal modes of EEG data derived from a multivariate Bayesian autoregressive function. DCM describes the observed CSDs by combining a biophysically plausible network model with a forward (observation) model that enables a mapping between the underlying network dynamics and the observed data. Here, we specified a network model consisting of five regions (cf. network architecture) that were each modeled as a neural mass (cf. neural mass model) and used the recommended boundary element model (BEM) for EEG data as a forward model. Each node of our network was treated as a patch on the cortical surface using the 'IMG' option (*Litvak et al., 2011*). The generative model can be inverted using a Bayesian inversion scheme, that is, the variational Laplace (*Friston et al., 2007*). Bayesian model inversion involves finding the posterior densities over model parameters that best explain the observed data by tuning the model parameters to minimize the discrepancy between the predicted and observed EEG data

under complexity constraints (i.e., deviations from prior values). This iterative model-fitting scheme provides one with posterior densities of the model parameters and a measure of the quality of the particular model (the log-evidence or free energy) that can be taken to the group level for further analysis. The CSD DCMs were inverted using the EEG data obtained during the ~120 s precision grip force task, covering the frequency range between 4 and 48 Hz. We chose this frequency range because both alpha, beta, and gamma oscillations have been implicated in mediating distinct sensorimotor control processes (*van Wijk et al., 2012*), and because the sensor data revealed prominent data features across different frequency bands (*Figure 1F*). Our data did not conform to the 1/f spectrum assumed by DCMs for CSD (originally developed for steady states, for example resting-state experiments). We therefore changed the prior of neural innovations (DCM.M.pE.a(2;:)) to −32 to assume a flat spectrum (*Spedden et al., 2020*). We further changed the hyperprior for the noise precision component (hE) from 8 to 18 to circumvent issues relating to early model convergence that was encountered during the initial attempts of model inversions. This issue is typically seen for non-linear electrophysiological DCMs (*van Wijk et al., 2018*; *Zeidman et al., 2019*). See the supplementary files (*Supplementary files 2* and *3*) for a presentation of our SPM and DCM pipelines.

## Network architecture

We fitted one full DCM at the subject level (first level). This included five sourcesin a uni-hemispheric sensorimotor network contralateral to the dominant hand: DLPFC, PMv, SMA, M1-hand, and IPL. This was motivated by the fact that these areas consistently have been implicated in visually guided pinch force control. Indeed, previous fMRI studies have found an increased BOLD response in these areas during comparable precision grasping tasks (*Ehrsson et al., 2001*; *Grol et al., 2007*). Reciprocal cortico-cortical connections between parietal and frontal regions provide the anatomical and functional basis for sensorimotor transformations. Specifically, it is largely accepted that higher sensory areas in the IPL are involved in integrating and processing multimodal sensory information. Some of this processing is directly useful for (grasping) actions, and information is conveyed to the PMv as a part of a grasping circuit (*Fogassi and Luppino, 2005*). Furthermore, SMA has also been attributed a role in fine control of force during precision grip tasks (*Kuhtz-Buschbeck et al., 2001*). M1 contains the majority of the corticomotoneuronal neurons that descend through the corticospinal pathway and are considered a main anatomical substrate for manual dexterous functions (*Lemon, 2008*) with specific roles in precision grips (*Muir and Lemon, 1983*). Increased activity in the DLPFC has also been demonstrated during precision grasping tasks (*Ehrsson et al., 2001*), and DLPFC displays a highly protracted developmental trajectory (*Gogtay et al., 2004*; *Shaw et al., 2008*). We opted for a uni-hemispheric network to avoid over-parameterizing the hypothesized network (DCMs for electrophysiological data are designed for modeling sparse network structures, and not for large numbers of sources that represent all possible regions active in the brain) and chose the contralateral hemisphere as brain activity (measured via fMRI) during online control of grasping action is more robust and larger in magnitude in the hemisphere contralateral to the hand used (*Cavina-Pratesi et al., 2018*; *Culham et al., 2006*; *Olivier et al., 2007*). One full model consisting of reciprocal connections between DLPFC-PMv, DLPFC-SMA, PMv-M1, SMA-M1, PMv-IPL, and SMA-IPL was fitted to each of the participants' EEG data (see *Figure 1*). As DCMs for electromagnetic data allow mapping of both forward ('driving'), backward ('modulatory'), and lateral ('general') connections, we specified the network with backward connections between DLPFC-PMv, DLPFC-SMA, PMv-M1, SMA-M1, PMv-IPL, and SMA-IPL and forward connections for the reverse direction. Reciprocal lateral connections were modeled between PMv-SMA as there is no evidence of a hierarchical difference between these regions (cf. below for a description of specificity of forward, backward, and lateral connections) (*Figure 1G*). These decisions were motivated by the proposed hierarchical organization of the sensorimotor system (*Adams et al., 2013*; *Felleman and Van Essen, 1991*; *Shipp, 2007*; *Shipp, 2005*; *Shipp et al., 2013*). We obtained MNI coordinates for the different nodes of interests from previous studies on brain activation during visually guided or sensory cued finger movements (*Figure 1E*; *Ehrsson et al., 2001*; *Ehrsson et al., 2000*; *Grol et al., 2007*). For left-handed individuals (n = 8), we flipped the MNI coordinates across the midline before estimating the DCMs to account for hand dominance.

## Neural mass model

Using DCMs for electromagnetic recordings enables fitting each node in the hypothesized network as a biophysically informed cortical column consisting of different subpopulations of neurons. Here, we opted for the convolution-based 'LFP' mass model (*Moran et al., 2013*; *Moran et al., 2009*), consisting of populations of excitatory and inhibitory neurons and pyramidal cells, as this was specifically developed to capture data features in the frequency domain. The output from each node to another is conveyed by the pyramidal cells targeting either the excitatory cell population (forward connections; physiologically assumed to exert 'driving' effects), the inhibitory cell populations and excitatory pyramidal cells (backward connections; physiologically assumed to primarily exert 'modulatory' effects), or all subpopulations of neurons, for example, for connections within the same level of hierarchy (lateral connections) (*Adams et al., 2013*; *Bastos et al., 2012*; *Moran et al., 2013*; *Shipp, 2005*; *Shipp et al., 2013*).

## Verifying network of included cortical sources

After having inverted one DCM per subject corresponding to the a priori defined network of interest, we also specified and inverted another DCM that resembled this network, but had the IPL modeled as a hidden source (i.e., without extrinsic connections to and from the IPL source). The purpose of this analysis was to determine whether the chosen network sources were sufficiently spaced as volume conduction of sources to the EEG may limit analyses of brain connectivity. The logic behind this analysis is that if a source contributes to the observed cross-spectra, this would be reflected in a difference in model evidence between the two models favoring the fully connected model (*full DCM*) over the disconnected model (*disconnected DCM*) (full DCM > disconnected DCM) (*Dietz et al., 2014*). We investigated this hypothesis using RFX BMS (*Rigoux et al., 2014*; *Stephan et al., 2009*). A RFX approach was favored over a fixed-effects (FFX) approach because it is likely that different individuals could display different model architectures (i.e., different magnitudes or influence of field spread). Here, we investigated the likelihood that the Full DCM provided a more parsimonious fit to the data than the Disconnected DCM among all subjects using exceedance probabilities and protected exceedance probability. These metrics quantify how likely it is that one model is more frequently represented within individuals than any other model(s) in the space of models compared (*Rigoux et al., 2014*; *Stephan et al., 2009*). The protected exceedance probability furthermore takes into account that differences in model evidence could be due to chance by marginalizing the exceedance probabilities (*Rigoux et al., 2014*).

## Group-level analysis using PEB

We next used the PEB framework to perform the group (second)-level analysis evaluating between-subject effects in connectivity from the winning DCM (i.e., the a priori full DCM) (*Friston et al., 2016*; *Zeidman et al., 2019*). PEB was used to infer commonalities in network architecture and coupling strength across individuals (mean connectivity) as well as differences between individuals due to age group; precision grip performance and interactions between age group and precision grip performance. These between-subject regressors of interest were entered in our PEB design matrix. This specified a Bayesian general linear model that partitioned between-subject variability into regressor effects and residual random effects. Parameter vectors are estimated from the data for every regressor in the design matrix (i.e., age group; precision grip performance and interactions between age group and precision grip performance). In general, these parameters can be interpreted as reflecting the group-level effect of one regressor on one DCM parameter. Notably, this feature allowed us to infer independent 'main effects' of age group and precision grip performance as well as interactions between the two. We chose to model age as a categorical variable (age group), and not as a continuous variable (for example age in months), as this was how participants were recruited. As our questions of interest were related to the extrinsic connectivity parameters, we took only values of inter-regional (extrinsic) connectivity to the second level of analysis (A-matrix; a total of 14 DCM parameters per individual). We allowed the between-subject regressors to influence all extrinsic connections. Next, we performed an exhaustive search of nested models, in which the Bayesian model evidence was evaluated and compared for all possible combinations of connections in the model space switched on or off (i.e., prior fixed at zero). This iterative process of BMR through the 'greedy search' scheme enabled pruning away connections that did not contribute

positively to the model evidence. This approach was motivated by our broad hypotheses (i.e., that we did not have strong claims regarding exactly where effects would be expressed) and was performed under two assumptions: (1) that all combinations of extrinsic coupling parameters were equally probable; and (2) that the full model only contained physiologically probable connection parameters (*Zeidman et al., 2019*). For our study, this approach was chosen because developmental differences in cortico-cortical connectivity during motor tasks are generally not well understood. An average of the models in the final iteration was then computed (BMA). This approach provided the following outcomes: mean (group) connectivity, the associated uncertainty and posterior probabilities across individuals (commonalities); differences in extrinsic coupling due to age group; due to differences in precision grip performance; and due to distinct relationships between performance and coupling strength because of age group, that is, interactions. This interpretative framework was enabled by mean-centering all regressors before they were entered in the PEB design matrix. To enable robust statistical inference, we thresholded averaged model parameters at >95% posterior probabilities (i.e., strong evidence of the parameters being present vs. absent). To visualize results, individual connectivity estimates for parameters of interest were extracted using the group-averaged connection strengths as priors. Coupling strength estimates from connections with >95% posterior probabilities were plotted against performance with boxplots displaying age group representations of coupling estimates. Of note, as we only included data from a single task condition, effects of age group are not necessarily task specific. See *Supplementary file 4* for a point-to-point presentation of the PEB analysis. We further extracted coupling estimates and subjected them to a LOOCV analysis using multiple linear regression to quantify how well coupling parameters performed in predicting out-of-sample fractionized age (measured in months) and precision grip performance. We also used a multiclass classifier – QDA – to predict which age group (as a categorical variable) participants belonged based on their estimated coupling strengths. These analyses were carried out using the *caret* R-package.

## Acknowledgements

We would like to thank the participants and their parents. MMB was funded by a grant from the Danish Ministry of Culture.

## Additional information

### Funding

| Funder | Grant reference number | Author |
|---|---|---|
| Ministry of Culture | FPK.2018-0070 | Mikkel Malling Beck<br>Jesper Lundbye-Jensen |
| Nordea-fonden | 02-2016-0213 | Jesper Lundbye-Jensen |

The funders had no role in study design, data collection and interpretation, or the decision to submit the work for publication.

### Author contributions

**Mikkel Malling Beck,** Conceptualization, Formal analysis, Investigation, Writing - original draft, Project administration; **Meaghan Elizabeth Spedden, Martin Jensen Dietz,** Formal analysis, Writing - review and editing; **Anke Ninija Karabanov,** Writing - review and editing; **Mark Schram Christensen,** Conceptualization, Supervision, Writing - review and editing; **Jesper Lundbye-Jensen,** Conceptualization, Supervision, Project administration, Writing - review and editing

### Author ORCIDs

Mikkel Malling Beck ![ORCID] https://orcid.org/0000-0002-8006-2580
Martin Jensen Dietz ![ORCID] https://orcid.org/0000-0003-0029-6932
Anke Ninija Karabanov ![ORCID] http://orcid.org/0000-0003-1874-393X

### Ethics

Human subjects: Before enrolling in the study, all participants assented to the study procedures and written informed consent was obtained from participants (> 18 years) and their parents (< 18 years). The study was approved by the regional ethical committee (protocol number: H-17019671).

### Decision letter and Author response

Decision letter https://doi.org/10.7554/eLife.61018.sa1
Author response https://doi.org/10.7554/eLife.61018.sa2

## Additional files

### Supplementary files

• Source code 1. MATLAB script used to preprocess the EEG data used for the dynamic causal modeling analysis.

• Supplementary file 1. Table representing the pipeline used in EEGLAB (v.14.1.1) interfaced in MATLAB R2017b.

• Supplementary file 2. Table representing steps applied during conversion of files to SPM data files in SPM12 (v7487) interfaced in MATLAB R2017b.

• Supplementary file 3. Table representing the pipeline used for estimating single-subject dynamic causal modelings in SPM12 (v7487).

• Supplementary file 4. Represents the pipeline used for the group-level (second level) analyses of dynamic causal modelings in SPM12 (v7487). *The parametric empirical Bayes analysis was run three times, with the design matrix reordered so that the main regressor of interest was listed just after the commonalities (column 2) as per https://en.wikibooks.org/wiki/SPM/Parametric_Empirical_Bayes_(PEB)#Search_over_nested_PEB_models.

• Transparent reporting form

### Data availability

Preprocessed data analyzed in this study has been made available on the Open Science Framework (https://osf.io/ap7ws/). Source data for Figure 1D, 4, 5 and 6 are provided.

The following dataset was generated:

| Author(s) | Year | Dataset title | Dataset URL | Database and Identifier |
|---|---|---|---|---|
| Beck MM | 2021 | Age-related differences in motor control | https://doi.org/10.17605/OSF.IO/AP7WS | Open Science Framework, 10.17605/OSF.IO/AP7WS |

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
