## [Decision Letter]

**Acceptance summary:**

Beck and colleagues present a study that provides a new perspective on how dexterous motor control is supported and refined across childhood and adulthood, of interest to researchers studying neuroscience, movement, and development. The authors use Dynamic Causal Modelling and Parametric Empirical Bayes to show that bidirectional coupling of EEG signals over parietal and premotor cortex is linked to precision force grip task performance from early childhood onwards, whilst contributions of other networks (including pre-motor and pre-frontal nodes) change into the teen-age years, potentially reflecting better top-down control. This work stands out in its use of model-based approaches with a large developmental sample to uncover how neural coupling dynamics of the canonical grasping network support precision grip performance.

**Decision letter after peer review:**

Thank you for submitting your article "Cortical signatures of precision grip force control in children, adolescents and adults" for consideration by *eLife*. Your article has been reviewed by 3 peer reviewers, one of whom is a member of our Board of Reviewing Editors, and the evaluation has been overseen by Tamar Makin as the Senior Editor.

As the editors have judged that your manuscript is of interest, but as described below that additional extensive analyses are required before it is published, we would like to draw your attention to changes in our revision policy that we have made in response to COVID-19 (https://elifesciences.org/articles/57162). First, because many researchers have temporarily lost access to the labs, we will give authors as much time as they need to submit revised manuscripts. We are also offering, if you choose, to post the manuscript to bioRxiv (if it is not already there) along with this decision letter and a formal designation that the manuscript is "in revision at *eLife*". Please let us know if you would like to pursue this option. (If your work is more suitable for medRxiv, you will need to post the preprint yourself, as the mechanisms for us to do so are still in development.)

Summary:

Beck and colleagues use a model-driven approach that links neural coupling dynamics in EEG to precision force grip task performance. Using DCM and PEB to characterise effective connectivity, they show that bidirectional coupling in the canonical 'grasping network' is associated with precision grip performance across childhood and adulthood. In addition, they link age-related changes in backward coupling from higher-order to lower-order sensorimotor regions to improved top-down control, providing a novel perspective on how motor control is refined across development.

Essential revisions:

All three reviewers of your manuscript agree that the empirical question this study addresses is interesting and timely, the manuscript is well-written, and the results are novel, very interesting, and have potential for enriching the field. All reviewers however, also had some significant concerns about the analyses and results, which they felt risk affecting the face validity of the conclusions. These are summarised below. We would like to offer you an opportunity to address these concerns and re-submit.

1) A key issue, raised by all reviewers, is the use of a restricted, unihemispheric network of sources. Particularly in a grasp-control task like this, homologous regions to the ones in the model have been found to be particularly important. This is especially true for the ipsilateral parietal regions. In addition, a paediatric fMRI study on grasp control development in children by Halder at al., (2007), suggests developmental changes in grasp control are linked to visual cortex function. Please explain clearly from the start why a unilateral network was used, and provide appropriate controls showing that conclusions are not substantially altered by including additional networks – or indeed, consider expanding analyses to include other task-relevant regions.

Halder, P., Brem, S., Bucher, K., Boujraf, S., Summers, P., Dietrich, T.,.… and Brandeis, D. (2007). Electrophysiological and hemodynamic evidence for late maturation of hand power grip and force control under visual feedback. Human brain mapping, 28(1), 69-84.

2) A second key concern is the close positioning of sources included in the DCM. The extremely close location of ROIs could result in significant confounds of signal spread, especially in EEG signals. How has this contributed to your findings? This needs to be addressed with adequate control analyses before publication in *eLife* can be considered.

3) Can we be sure DCM estimates are sufficiently robust to predict within age-group performance differences? The interactions with age and performance are under-powered compared to the main effects due to smaller cell size. In particular, the flip in the performance-coupling relationship for SMA-M1 coupling depending on age seems unexpected, raising questions about whether these results should be interpreted with some caution. Please address this issue, for example by considering using age as a continuous rather than a grouping variable , or ensuring that performance can also be predicted from DCM parameters within age group.

4) Relatedly, the discussion from line 329, implies a link between performance, backward neural coupling parameters, and age, but cites weak evidence for such a link in this dataset. This interpretation also seems at odds with the conclusion on lines 321-323 that performance-relevant network dynamics develop early. To avoid confusion, please amend text to signpost clearly between task-relevant and task-independent age effects, and between interpretations based directly on significant results versus broader speculations.

[Editors' note: further revisions were suggested prior to acceptance, as described below.]

Thank you for resubmitting your work entitled "Cortical signatures of precision grip force control in children, adolescents and adults" for further consideration by *eLife*. Your revised article has been evaluated by Tamar Makin (Senior Editor) and a Reviewing Editor.

The manuscript has been improved and all queries regarding potential confounds in the DCM analyses have been addressed. This leaves only one important point concerning the specificity of overall age differences in EEG and DCM for precision grasping that requires further clarification before we can accept this manuscript for publication:

In several places in the original version of the manuscript, you interpreted main effects of age as developmental change in grasp control-relevant processes. Across several comments in the review it was suggested these differences may also reflect task-irrelevant factors such as changing scalp dynamics, signal amplitude, noise, etc. While you generally agreed with these comments and have toned down some wording in response, this issue is still not openly discussed in the current version of the manuscript. Without baseline task or other control for age-related confounds, the relevance of simple age effects for the development of grasping is questionable. Please provide a transparent reflection on this major confound, and a rebalancing of the main conclusions in the light of this issue.

---

## [Author Response]

Essential revisions:All three reviewers of your manuscript agree that the empirical question this study addresses is interesting and timely, the manuscript is well-written, and the results are novel, very interesting, and have potential for enriching the field. All reviewers however, also had some significant concerns about the analyses and results, which they felt risk affecting the face validity of the conclusions. These are summarised below. We would like to offer you an opportunity to address these concerns and re-submit.1) A key issue, raised by all reviewers, is the use of a restricted, unihemispheric network of sources. Particularly in a grasp-control task like this, homologous regions to the ones in the model have been found to be particularly important. This is especially true for the ipsilateral parietal regions. In addition, a paediatric fMRI study on grasp control development in children by Halder at al., (2007), suggests developmental changes in grasp control are linked to visual cortex function. Please explain clearly from the start why a unilateral network was used, and provide appropriate controls showing that conclusions are not substantially altered by including additional networks – or indeed, consider expanding analyses to include other task-relevant regions.Halder, P., Brem, S., Bucher, K., Boujraf, S., Summers, P., Dietrich, T.,.… and Brandeis, D. (2007). Electrophysiological and hemodynamic evidence for late maturation of hand power grip and force control under visual feedback. Human brain mapping, 28(1), 69-84.

We thank the reviewers for giving us the chance to elaborate on this matter. First, we would like to stress that the aim of our study was to test hypotheses about age and performance-specific changes in the extrinsic connectivity between key regions of a uni-hemispheric contralateral grasping network defined a priori. Our goal was not to test hypotheses about the number of regions active during our task or to test hypotheses about which model architecture provided the best explanation of the observed data for different age groups. One key advantage of the chosen approach was that we were able to investigate how estimated connectivity associated with motor performance measures (continuous data) (Zeidman et al., 2019). In doing so, we had to restrict our model to feature sources consistently reported to be involved in precision grip control and/or sources in brain regions displaying clear developmental changes from childhood to adolescence. In general, DCM for electrophysiological data is designed for modelling sparse network structures and not well suited to represent large numbers of sources as complex models are known to increase the chances of local minima in the free energy objective function during DCM estimation (Litvak et al., 2019; Zeidman et al., 2019). Therefore, we focused on sources in the contralateral hemisphere that has consistently been implicated in the control of a precision grip.

We do however acknowledge that the interpretations of our results are contingent upon the structure of the model(s) defined and tested. We have therefore added the following sentence to the discussion (p. 16, line 452-459) to highlight this challenge:

“A consequence of this is that models with different connectivity architectures might offer equally – or even more – probable explanations of the observed data. Of note, DCM for electrophysiological data is designed for modelling sparse network structures and do not accommodate large numbers of sources well as this may increase the risk of local minima during model inversion (Litvak et al., 2019; Zeidman et al., 2019). We chose to fit our DCM to cortical sources consistently implicated in motor and executive control processes, and restricted the network to the contralateral hemisphere, as this has most robustly been implicated in control of the precision grip (e.g. Cavina-Pratesi et al., 2018; Culham et al., 2006).”

We have also added the following to the method section in which we elaborate on our reasons for restricting ourselves to the contralateral grasping network:

“We opted for a uni-hemispheric network to avoid over-parameterizing the hypothesized network (DCMs for electrophysiological data are designed for modelling sparse network structures, and not for large numbers of sources that represent all possible regions active in the brain) and chose the contralateral hemisphere, as brain activity (measured via fMRI) during online control of grasping action is more robust and larger in magnitude in the hemisphere contralateral to the hand used (Cavina-Pratesi et al., 2018; Culham et al., 2006; Olivier et al., 2007).” (p. 20, line 564-569)

A more detailed review of the imaging literature that underlines the core role of contralateral sources of the grasping network can be found below. In brief, we identified neuroimaging studies with human participants performing a precision grip. We identified 41 studies where the majority involved adult individuals. One study investigated brain activation related to precision grip in children (4-6 years).

The literature review shows that the majority of performed studies find evidence of contralateral parietal activation (in regions close to the inferior parietal lobule, IPL). Specifically, this was the case for 39/41 of the identified studies (cf. Table 1, attached). In contrast, 26/41 found evidence of ipsilateral IPL response. Furthermore, in several of the studies reporting ipsilateral parietal activation, the authors report a greater and more consistent contralateral than ipsilateral activation. The review of this literature also outlines why we chose not to include the visual cortex. Indeed, only a few studies (14/41) have reported increased BOLD responses in visual regions while performing a precision grip (Table 2, attached). Furthermore, we could not find evidence of specific brain regions consistently reported to display increased activation.

Altogether, we believe that this summary supports our modelling approach with focus on the core nodes of the contralateral grasping network.

2) A second key concern is the close positioning of sources included in the DCM. The extremely close location of ROIs could result in significant confounds of signal spread, especially in EEG signals. How has this contributed to your findings? This needs to be addressed with adequate control analyses before publication in eLife can be considered.

We agree with the reviewers that the close location of sources included could have confounded the results of the DCM through signal spread of EEG data due to volume conduction. We have therefore now added results from the following analysis to the main body of the manuscript that investigates this question. In this analysis we compare the evidence of two different DCMs in all participants: one full model and one in which the contralateral IPL was modelled as a hidden source. The rationale behind such an analysis is that if a source contributes to explaining the observed cross spectra, this would be reflected in a difference in model evidence between the two models favoring the fully connected model (full DCM) from the disconnected model (Disconnected DCM) (m1 > m2). We find that the fully connected DCM is favored over the disconnected, and believe that this provides strong support that volume conduction has not substantially contaminated the results from the full model.

We addressed the potential role of volume conduction in our a priori defined network. First, we defined two different DCMs: One full DCM (Full DCM) with all sources fully connected (as in the a priori defined model used in the original version of the manuscript, see Figure 1F and Figure 3) versus another DCM without extrinsic connections from the inferior parietal lobule (IPL) to and from PMv and SMA (Disconnected DCM) (see Figure 3). IPL was chosen as the source of interest to be modelled for the given question, as this was close in proximity to M1 (note the MNI coordinates, see Figure 1, Figure 3 and Figure 2—figure supplement 1). The rationale behind such an analysis is that if a source contributes to explaining the observed cross spectra, this would be reflected in a difference in model evidence between the two models favoring the fully connected model over the disconnected model (Full DCM > Disconnected DCM).

We investigated this hypothesis using random-effects (RFX) Bayesian model selection (BMS) (Stephan et al., 2009; Rigoux et al., 2014). A random-effects approach was favored over a fixed-effects approach (FFX) because it is likely that different individuals could display different model architectures (i.e. different magnitudes of signal spread). Here, we investigated the likelihood that the Full DCM was more frequent than the Disconnected DCM among all subjects (n=88) using the exceedance probabilities and protected exceedance probability. These metrics quantify how likely it is that one model is more frequently represented within individuals than any other model(s) in the space of models compared (Stephan et al., 2009; Rigoux et al., 2014). The protected exceedance probability furthermore takes into account that differences in model evidence could be due to chance by marginalizing the exceedance probabilities (Rigoux et al., 2014).

Using RFX BMS we found that the exceedance probability of the Full DCM was 1 whereas that of the Disconnected DCM was 0 (Figure 3 in the revised manuscript). The protected exceedance probability was 0.99 for the Full DCM and 0.01 for the Disconnected DCM (see figure 3), and the probability of equal model frequencies (BOR) was < 0.01. It is therefore highly probable that the full DCM provides a better fit to the data across all participants. This favors the view that keeping the IPL source (the sources closest in proximity to the M1 source) provides a more parsimonious fit to the data under the free-energy approximation to the Bayesian model evidence. Note that the Bayesian model evidence not only scored the additional variance explained by the IPL source, but also penalizes the extra complexity incurred by adding extra parameters. As such, the IPL source explains a unique proportion of variance that would not be the case, if this was heavily influenced by signal spread of the EEG. Given that the IPL-M1 sources were the spatially closest sources we argue that it is very unlikely that volume conduction would contaminate our results in general.

The following paragraph has been added to the Results section of the manuscript:

“Verifying network of included cortical sources

Before moving on to estimating age-related differences in effective brain connectivity between individuals on the group level, we set out to verify that the sources included in the DCM were sufficiently spaced, as volume conduction can affect measures of cortico-cortical connectivity in EEG data (Schoffelen and Gross, 2009). […]It therefore seems likely that each of the sources included in our DCM contribute uniquely to explaining the observed data features.”

Furthermore, we have added the following paragraph to the Method section to describe the rationale and the approach:

“Verifying network of included cortical sources

After having inverted one DCM per subject corresponding to the a priori defined network of interest network of interest, we also specified and inverted another DCM that resembled this network, but had the IPL modelled as a hidden source (i.e. without extrinsic connections to and from the IPL source). […] The protected exceedance probability furthermore takes into account that differences in model evidence could be due to chance by marginalizing the exceedance probabilities (Rigoux et al., 2014).”

3) Can we be sure DCM estimates are sufficiently robust to predict within age-group performance differences? The interactions with age and performance are under-powered compared to the main effects due to smaller cell size. In particular, the flip in the performance-coupling relationship for SMA-M1 coupling depending on age seems unexpected, raising questions about whether these results should be interpreted with some caution. Please address this issue, for example by considering using age as a continuous rather than a grouping variable , or ensuring that performance can also be predicted from DCM parameters within age group.

We thank the reviewers for giving us the chance to elaborate on this subject.

Age was treated as a categorical variable rather than a continuous variable because this was true to the design of the experiment. Indeed, we recruited participants from four different age groups and not continuously across the age span from 8-30 years. A consequence of this is that even though one considers age as a continuous variable (e.g. age in months), individuals would still be ‘grouped’ within fairly narrow ranges of ages with rather large ‘jumps’ between clusters of data points. Consequently, treating age as a continuous variable could lead to spurious associations to coupling strength, i.e. associations that are apparent when analyzed globally (across clusters), but not there (or even have different signs) when analyzed within each group (cluster) (see e.g. figure 4 and 6 in Kievit et al., 2013 and also Makin and De Xivry, 2019). This is also known as the Simpson’s paradox. We have added the following paragraph to the manuscript to highlight this choice:

“We chose to model age as a categorical variable (age group), and not as a continuous variable (e.g. age in months), as this was how participants were recruited.”

Additionally, although we fully agree that splitting data into categories could lead to a loss in power when data is originally sampled on a continuous scale (Fedorov et al., 2009), we do not think that this is relevant for the data collected in the present study due to the nature of recruitment of participants.

Furthermore, the parametric empirical Bayes (PEB) analysis can be considered as a Bayesian general linear model that includes a number of regressors (Zeidman et al., 2019). In the present study, we included regressors for ‘Performance’, ‘Age group’ and ‘Performance x Age group’. Given that all regressors were included in one GLM, we can consider the effects of each variable of interest (here, ‘Performance’) to be independent of the others (here ‘Age group’ and ‘Performance x Age group’ interactions). That is, parameters encode the differences between participants due to each regressor. Therefore, we argue that DCM-PEB parameters are indeed capable of detecting relationships between connectivity strength and precision grip performance that are independent of differences due to age group. We acknowledge that this was not clear in the original version of the manuscript. The following explanation has therefore been added to the methods section (p. 22, line 625-630):

“Parameter vectors are estimated from the data for every regressor in the design matrix (i.e. age group; precision grip performance and interactions between age group and precision grip performance). In general, these parameters can be interpreted as reflecting the group-level effect of one regressor on one DCM parameter. Notably, this feature allowed us to infer independent ‘main effects’ of age group and precision grip performance as well as interactions between the two”.

The results presented in Figure 5AB also demonstrate that performance can indeed be predicted within groups. The correlation coefficients describing the linear association between estimated coupling strengths and motor performance are in the same direction for all groups in all but one case (7/8). We have added the following sentence to the manuscript to highlight this feature (p. 9, line 235-236).

“Notably, these relationships were consistently present across different age groups as can be seen from the group-wise Pearson correlation coefficients presented in Figure 5AB.”

4) Relatedly, the discussion from line 329, implies a link between performance, backward neural coupling parameters, and age, but cites weak evidence for such a link in this dataset. This interpretation also seems at odds with the conclusion on lines 321-323 that performance-relevant network dynamics develop early. To avoid confusion, please amend text to signpost clearly between task-relevant and task-independent age effects, and between interpretations based directly on significant results versus broader speculations.

Thank you for bringing this to our attention. We agree with the reviewers that parts of the discussion lacked conciseness. We have amended the text throughout the discussion to clearly distinguish between task-relevant and task-independent effects; and between interpretations based on empirical evidence from the present study and general speculations based on results from other studies. For example, the following changes have been made:

“The results of the present study indicate that backward connectivity from SMA to IPL was positively related with age. Given that backward connections mediate top-down control, this result suggests that, independent of task-performance, individuals aged 16 and above display a larger degree of top-down control. Additionally, for the age group-performance interactions, the estimated backward coupling from SMA to M1 only followed a clear negative trend with performance for individuals younger than 14, possibly reflecting an inability to capitalize on efficient top-down control to guide precision grip performance for younger individuals.” (p. 14, line 371-379).

In addition, on page 14 line 380-387, we have removed sentences, in which we refer to results from a previous study from our group, as this could have caused confusion:

“This suggestion of a greater degree of efficient top-down control with advancing age harmonize well with a previous study from our group investigating oscillatory coupling within the corticospinal system. Using directionality analyses of corticomuscular coherence, we demonstrated a larger proportion of descending, relative to ascending, oscillatory coupling as a function of age (in 7-23 y.o. participants) during a force tracing task using the ankle muscles (Spedden et al., 2019). Echoing the results from the current study, the greater proportion of descending relative to ascending oscillatory coupling could support the notion of an increased reliance on predictive, top-down control with age.”

Other changes to the Discussion section have been tracked as changes in the revised version of the manuscript. We would like to thank the reviewers for highlighting this shortcoming, and we believe the amendments that we have made have improved the conciseness of the discussion.

References:

Arnal LH, Giraud AL. 2012. Cortical oscillations and sensory predictions. Trends Cogn Sci.

Arnal LH, Wyart V, Giraud AL. 2011. Transitions in neural oscillations reflect prediction errors generated in audiovisual speech. Nat Neurosci. 14:797–801.

Bonaiuto JJ, Meyer SS, Little S, Rossiter H, Callaghan MF, Dick F, Barnes GR, Bestmann S. 2018. Lamina-specific cortical dynamics in human visual and sensorimotor cortices. *eLife*. 7.

Cavina-Pratesi C, Connolly JD, Monaco S, Figley TD, Milner AD, Schenk T, Culham JC. 2018. Human neuroimaging reveals the subcomponents of grasping, reaching and pointing actions. Cortex. 98:128–148.

Cragg L, Kovacevic N, Mcintosh AR, Poulsen C, Martinu K, Leonard G, Paus T. 2011. Maturation of EEG power spectra in early adolescence: A longitudinal study. Dev Sci. 14:935–943.

Culham JC, Cavina-Pratesi C, Singhal A. 2006. The role of parietal cortex in visuomotor control: What have we learned from neuroimaging? Neuropsychologia. 44:2668–2684.

Fedorov V, Mannino F, Zhang R. 2009. Consequences of dichotomization. Pharm Stat. 8:50–61.

Kievit RA, Frankenhuis WE, Waldorp LJ, Borsboom D. 2013. Simpson’s paradox in psychological science: a practical guide. Front Psychol. 4:513.

Litvak V, Jafarian A, Zeidman P, Tibon R, Henson RN, Friston K. 2019. There’s no such thing as a “true” model: The challenge of assessing face validity. In: Conference Proceedings – IEEE International Conference on Systems, Man and Cybernetics. Institute of Electrical and Electronics Engineers Inc. p. 4403–4408.

Makin TR, De Xivry JJO. 2019. Ten common statistical mistakes to watch out for when writing or reviewing a manuscript. *eLife*. 8.

Miskovic V, Ma X, Chou CA, Fan M, Owens M, Sayama H, Gibb BE. 2015. Developmental changes in spontaneous electrocortical activity and network organization from early to late childhood. Neuroimage. 118:237–247.

Moran RJ, Stephan KE, Seidenbecher T, Pape HC, Dolan RJ, Friston KJ. 2009. Dynamic causal models of steady-state responses. Neuroimage. 44:796–811.

Rigoux L, Stephan KE, Friston KJ, Daunizeau J. 2014. Bayesian model selection for group studies – Revisited. Neuroimage. 84:971–985.

Rodríguez-Martínez EI, Barriga-Paulino CI, Rojas-Benjumea MA, Gómez CM. 2014. Co-Maturation of Theta and Low-β Rhythms During Child Development. Brain Topogr. 28:250–260.

Spedden ME, Jensen P, Terkildsen CU, Jensen NJ, Halliday DM, Lundbye-Jensen J, Nielsen JB, Geertsen SS. 2019. The development of functional and directed corticomuscular connectivity during tonic ankle muscle contraction across childhood and adolescence. Neuroimage. 191:350–360.

Stephan KE, Penny WD, Daunizeau J, Moran RJ, Friston KJ. 2009. Bayesian model selection for group studies. Neuroimage. 46:1004–1017.

van Wijk BCM, Beek PJ, Daffertshofer A. 2012. Neural synchrony within the motor system: What have we learned so far? Front Hum Neurosci.

Wang XJ. 2010. Neurophysiological and computational principles of cortical rhythms in cognition. Physiol Rev.

Zeidman P, Jafarian A, Seghier ML, Litvak V, Cagnan H, Price CJ, Friston KJ. 2019. A guide to group effective connectivity analysis, part 2: Second level analysis with PEB. Neuroimage. 200:12–25.

[Editors' note: further revisions were suggested prior to acceptance, as described below.]

In several places in the original version of the manuscript, you interpreted main effects of age as developmental change in grasp control-relevant processes. Across several comments in the review it was suggested these differences may also reflect task-irrelevant factors such as changing scalp dynamics, signal amplitude, noise, etc. While you generally agreed with these comments and have toned down some wording in response, this issue is still not openly discussed in the current version of the manuscript. Without baseline task or other control for age-related confounds, the relevance of simple age effects for the development of grasping is questionable. Please provide a transparent reflection on this major confound, and a rebalancing of the main conclusions in the light of this issue.

We thank the senior editor and the reviewing editor for giving us the chance to submit our revised manuscript. We agree that task-irrelevant factors could, in principle, have affected the results and we have discussed this issue in greater detail in the current revision.

Below, we also highlight the changes we have made in the manuscript in the current revision. The following changes have been made:

1) The initial part of the discussion in which we summarize main findings has been revised to avoid ambiguity between reported effects that are strictly performance-related; those that address age-dependent differences; and those that reflect the interaction between the two:

“Additionally, independent of performance, individuals aged 16 and above expressed a greater degree of backward coupling between supplementary motor and parietal regions while a positive association between premotor and prefrontal coupling strength and performance was also selectively found in these individuals. This potentially reflects a greater degree of top-down control and utilization of additional executive control processes with positive implications for precision motor control from late-adolescence and onwards compared to childhood. Below, we discuss these patterns of task-based cortical connectivity. We focus on functionally relevant connectivity estimates that are related to performance across individuals independent of age; those that differ due to age group, but are not specifically related to motor performance; and those that relate to performance distinctively as a function of age.” (p. 13, line 313-322).

2) The following sentences have been added to the section discussing age-dependent effects to highlight that certain patterns of connectivity reflect general age-related differences, but are not related to development of skilled precision grip performance:

“Instead, our results suggest that some degree of functional refinement of connectivity within sensorimotor circuits extends into adolescence only reaching adult-like properties late during adolescence and that these differences are revealed during task-relevant processing. Although analyses were based on EEG recordings that were obtained while participants performed a precision grip task, these specific developmental differences in cortical connectivity were not related to precision grip performance per se as revealed by the PEB analysis. The functional relevance of the reported differences in coupling may therefore not be specific to the development of skilled grasping. Instead, they could reflect a general reorganization of directed brain connectivity patterns that may also be seen for other behavioral states or tasks. This also resonates well with previous findings suggesting a shift from bottom-up to top-down processing during cognitive tasks (e.g. taxing response inhibition) as children and adolescents approach adulthood (Bitan et al., 2006; Hwang et al., 2010).” (p. 15, line 391-396).

3) In the end of the section discussing age-dependent effects, we have now added a discussion on potential methodological confounds related to these aspects of the manuscript. This has been done to discuss the confounds related to the lack of a control condition:

“Collectively, this suggests that greater reliance on top-down processing at the expense of bottom-up processing with advancing age may reflect a common neurodevelopmental hallmark for functional brain interactions. The design of the study did not include a control task or control state (e.g. a recording during rest), and we therefore cannot make direct comparisons between task-specific and general differences in connectivity across ages. This also means that the observed age-related differences could be influenced by task-irrelevant factors such as general differences in the signal-to-noise ratio and how well the DCMs captured the oscillatory patterns of the obtained EEG signals. That being said, our primary interpretations are based on estimated connectivity obtained through inversion of DCMs, and we did not observe statistical differences in how well the estimated DCMs explained the variance of the EEG data features between age groups.” (p. 15, line 405-412).

4) In the final paragraph of the manuscript summarizing our results, we have now rebalanced the text to specify that we present results that reflect both age-dependent (independent of performance) and performance-dependent (independent of age) connectivity as well as interactions between the two.

“Collectively, our results link measures of directed brain connectivity within an extended sensorimotor network to skilled motor behavior in a precision grip task in children, adolescents and adults. In doing so, the results expand our knowledge on the cortical control mechanisms that support dexterous motor functions in humans and their development from childhood to adulthood.

*“*Collectivity, our results provide a characterization of differences in effective brain connectivity within an extended cortical sensorimotor network related to age and performance in a precision grip motor task in children, adolescents and adults. In doing so, the results expand our knowledge on the cortical control mechanisms that support dexterous motor functions in humans and how performance-specific and general patterns of connectivity develop from childhood to adulthood*.”* (p. 17, line 468-472).